# The Impact of Reward Object on Object-Based Attention

**DOI:** 10.3390/bs14060505

**Published:** 2024-06-18

**Authors:** Feiyu Diao, Xiaoqian Hu, Tingkang Zhang, Yunfei Gao, Jing Zhou, Feng Kong, Jingjing Zhao

**Affiliations:** 1School of Psychology, Shaanxi Normal University, Xi’an 710062, China; diao9741@snnu.edu.cn (F.D.); huxiaoqian@snnu.edu.cn (X.H.); ztk1874@snnu.edu.cn (T.Z.); gyyunfei@snnu.edu.cn (Y.G.); zhoujing.z@snnu.edu.cn (J.Z.); 2Shaanxi Provincial Key Laboratory of Behavior and Cognitive Neuroscience, Xi’an 710062, China

**Keywords:** reward object, selective attention, space-based attention, object-based attention

## Abstract

Reward has been shown to influence selective attention, yet previous research has primarily focused on rewards associated with specific locations or features, with limited investigation into the impact of a reward object on object-based attention (OBA). Therefore, it remains unclear whether objects previously associated with rewards affect OBA. To address this issue, we conducted two experiments using a paradigm that combined a reward training phase with a modified two-rectangle paradigm. The results indicate that a reward object modulates both space-based attention (SBA) and OBA. When cues appear on a reward object, the effects of both SBA and OBA are amplified compared to when cues appear on a no-reward object. This finding supports the value-driven attentional capture (VDAC) theory, which suggests that a reward object gain enhanced saliency to capture attention, thereby providing a theoretical support for the treatment of conditions such as drug addiction.

## 1. Introduction

Selective attention is crucial for human survival in a complex external world filled with various stimuli. Researchers have shown significant interest in this ability [1]. Recent research has revealed that selective attention operates not only based on location, akin to a searchlight or zoom lens (referred to as space-based attention, SBA) [2,3], but also is constrained by the boundaries of objects (i.e., object-based attention, OBA) [4,5]. When attention is focused on an object, locations within the object receive greater attention compared to locations outside the object. Extensive research supports the phenomenon known as the OBA effect [6,7,8].

Furthermore, some objects are more deserving of attention than others. Studies have revealed that stimuli that are frequently associated with rewards tend to capture more attention when they reappear [9,10]. For instance, if you win a lottery jackpot with a specific number, you are likely to pay special attention to this number in your future experiences. However, it remains unclear whether rewarded objects exert different constraints on attention compared to no-reward objects. Therefore, this study is designed to investigate whether reward objects influence OBA through two experiments.

### 1.1. Related Research

To investigate the influences on OBA, the two-rectangle paradigm developed by Egly et al. [11] has been extensively employed by previous researchers. This paradigm presents two parallel rectangles and a central fixation point, followed by the random appearance of a cue on one side of the rectangles. The target may appear in the same location as this cue (valid condition), on the other side of the cued rectangle (invalid same-object condition, ISO), or on the uncued rectangle (invalid different-object condition, IDO). Fastest responses to cued locations reflect the presence of the SBA effect, while faster responses to ISO compared to IDO reflect the manifestation of object advantage. Therefore, in the research reported here, we adapted a modified two-rectangle paradigm that incorporated the factor of reward object, thereby investigating the respective effects of a reward object on SBA and OBA.

By substituting traditional two-rectangle paradigm with high- and low-value money objects, Zhao et al. [12] investigated the association of currency with the OBA effect. They found a notable increase in the OBA effect when cues were presented on high-value bills compared to low-value ones. However, there are still some unresolved issues in their research. Firstly, monetary objects possess a multitude of physical attributes unrelated to rewards but may influence the OBA effect. Previous research has demonstrated that factors such as color, familiarity, long-term selection experience and the existence of a face [4,13,14,15] can influence OBA effects. Currency represents a complex object resulting from the integration of these attributes. Although Zhao et al. considered the influence of color factors, the impact of other factors and the inherent complexity of the objects themselves render monetary objects unable to be considered as purely reward objects. Secondly, Zhao et al.’s study used high- and low-value currency in their experiments, with the results dependent on comparisons between high–low denomination feature relationships, which are distinct from the attentional capture effects inherent in the reward object [16,17]. Lastly, reward objects refer to objects previously associated with reward acquisition. Since currency carries too many functions in daily life, it is reasonable to assume that in order to investigate whether reward objects affect selective attention, a purer form of reward object was required to investigate whether reward-associated objects produce special attentional effects compared to no-reward objects.

### 1.2. The Current Research

Our main hypothesis regarding the research results is grounded in the theoretical support from previous studies concerning the correlation between reward and selective attention. Anderson et al. [9] showed that information previously linked to high rewards tends to capture more selective attention compared to stimuli associated with no rewards, a concept consolidated in the subsequent theory of value-driven attentional capture (VDAC). This theory posits that stimuli associated with rewards gain additional salience, making them more likely to capture attention. Similarly, a number of researchers have found that when distractors are reward-relevant, they tend to induce strong cognitive interference due to their attentional attraction [18]. Based on the aforementioned research, a reward object is expected to capture attention more strongly, thereby modulating OBA effects. Specifically, we hypothesized that when cues appear on a reward object, participants would have greater difficulty in shifting their attention to another object, whereas when cues appear on a no-reward object, participants would exhibit faster attentional shifts to another object.

Another noteworthy point is that, according to the VDAC theory, when a cue appears on a reward object, that location should possess the strongest salience resulting from the combined effects of the cue and the reward object, thus leading to faster responses and increasing SBA effects compared to when the cue appears on a no-reward object [19]. Examining the influence of the reward object on SBA may support the universality of the VDAC theory while extending existing research on the distinction of the reward object from other types of reward factors. Therefore, our second hypothesis is that a reward object similarly modulates SBA effects.

To verify these two hypotheses, we presented data from two experiments in which the rectangles (Experiment 1) and the real objects (Experiment 2) were used to verify the stability of the experimental results. The only difference between Experiment 1 and Experiment 2 was the experimental materials used. This was conducted to further investigate social interactions: we are confronted with more complex objects and stimuli, and whether this effect remains stable when the stimulus’s complexity increases is yet to be examined. We explored how a reward object impacts SBA and OBA using a two-rectangle paradigm that combined a reward training phase. The reward training phase aimed to establish a connection between specific objects and rewards, with participants naïve to this link, allowing it to function independently as a reward object. More importantly, in the subsequent testing phase, we replaced the rectangle in the traditional two-rectangle paradigm with a rectangle that had just been rewarded, thereby investigating the impact of a reward object on SBA and OBA. If according to the VDAC theory reward objects influence subsequent attentional selection by acquiring larger saliency, then both OBA effects and SBA effects should be modulated by the reward object, as we previously hypothesized.

## 2. Method

### 2.1. Participants

To calculate the required sample size, we inputted the estimated effect size of reward object into a simulated two-way (cue position × cue validity) repeated-measures analysis of variance (ANOVA). Using G*Power 3.1 [20], we conducted this analysis. The power analysis, with an alpha level of 0.05, indicated that at least 28 participants would yield a power of 0.90. This sample size was determined to detect a medium effect (f = 0.25) [21].

Considering the calculation results of G*Power and the sample size of relevant studies [12], a total of 72 participants were used in this experiment. Of these, 36 participants took part in Experiment 1 (25 females, 11 males, age: M = 20 years, SD = 2.89) and another 36 participants took part in Experiment 2 (20 females, 16 males, M = 19.72 years, SD = 1.43). Every participant gave informed consent, had normal or corrected-to-normal visual acuity, and was naive to the experimental intent.

### 2.2. Apparatus and Stimuli

The Stimuli were displayed on a 19-in. color monitor with a resolution of 1280 × 1024 pixels and a refresh rate of 60 Hz. Participants were seated approximately 25 inches away from the monitor. In the training phase, a central fixation point was used, consisting of a plus sign measuring 0.48° × 0.48°, presented on a black background. The fixation point was followed by a colored rectangle. In Experiment 1, the rectangle was colored in either red (R = 194, G = 39, B = 39) or orange (R = 185, G = 134, B = 0), which had been checked and calibrated to the same brightness. In Experiment 2, real objects replaced the colored rectangles (as shown in the Figure 1). The real objects were umbrellas and doors, which were the same stimuli as used in Song et al. [22]. Previous studies have demonstrated no differences in terms of eliciting object-based attention and familiarity between these stimuli. In the test phase, the same fixation point was flanked by two different rectangles (2° × 4.9°) on the black background.

### 2.3. Design and Procedure

Both experiments were divided into a training phase and a test phase. Participants were initially trained to establish the association between rewards and colors during the training phase. Subsequently, we evaluated the impact of this reward object on selective attention during the test phase. In both the training and test phases, participants began with a practice block and proceeded to the formal experiment once they achieved an accuracy rate of 90%.

The task during the training phase involved a color-judgement task. In particular, each trial initiated with a central fixation lasting 500 ms, followed by the presentation of a target rectangle for 1500 ms or until the participant responded. The task of participants was to discern the color of the target rectangle by pressing ‘J’ key for red and ‘K’ key for orange rectangle. Rectangles with one color were probably rewarded (80% gained 0.05 RMB per trial and 20% gained no rewards) and rectangles of another color gained no rewards. The reward color was counterbalanced across participants, while the rectangle’s orientation (vertical or horizontal) was counterbalanced within participants. After the display of target, a 1000 ms feedback was followed and the next trial began; the feedback interface displayed whether the participant’s payment would increase based on the participant’s response. Upon completing the experiment, participants would receive extra rewards based on the final cumulative monetary earnings, with a maximum of 10 RMB. During the training phase, each participant completed a total of five blocks, each containing 100 trials. A resting interface was presented after each block, allowing participants to view their current cumulative additional rewards obtained up to that point (see Figure 2a).

In the test phase, participants were instructed to complete a modified probe task based on the two-rectangle paradigm. A total of 640 trials were conducted, consisting of 512 target-present trials and 128 no-target trials. Among trials where probe was presented, 256 (50%) were reserved for the valid condition. The remaining 256 were evenly divided between ISO condition and IDO condition. And, each trail commenced with a display that included a fixation cross and two rectangles, one with rewarded color and another with no-rewarded color in the training phase; the position of these two rectangles was counterbalanced within participants. A cue was presented at one of the four corners of the two rectangles at random for 100 ms after a 1000 ms interval. This was succeeded by a 200 ms absent stimulus gap. Subsequently, the target was presented (or no stimulus for no-target trials, in the case of target-absent catch trials) and stayed on the screen for 1500 ms or the duration of the participant’s response (refer to Figure 2b). The subsequent trial commenced following a 500 ms intertrial pause, during which the monitor remained empty. Participants were directed to fixate on the central point throughout the trial and to respond promptly upon detecting a target. The experiment comprised of four blocks, consisting of 160 trials per block interspersed with a rest period between them.

## 3. Result

### 3.1. Training Phase

In the training phase of both Experiment 1 and Experiment 2, we trained the subjects to establish associations between different rectangles and the presence or absence of rewards. Figure 3a,b shows the observed RTs differences during this phase, which we used as an indicator of whether the associations had been established. The RTs for reward condition were faster by 16 ms compared to those for no-reward condition (t = 3.95, *p* < 0.001, Cohen’s d = 0.66) in Experiment 1, which meant that the associations were successfully established. In the training phase of Experiment 2, a similar *t*-test for reward condition was executed and showed that the RTs for reward trials were 22 ms faster than those for no-reward trials (t = −4.975, *p* < 0.001, Cohen’s d = −0.83, see Figure 3d), which was consistent with the results from Experiment 1.

### 3.2. Test Phase

Correct responses were identified as keypresses made between three standard deviations from the mean following the target’s onset. RTs falling outside of this specified range were excluded, constituting 0.2% for unsuccessful keypresses and 2.1% for the use of three standard deviations. In Experiment 2, the proportion of excluded participants was 1.3%.

#### 3.2.1. Experiment 1

Given the success of these established associations, we tested in the testing phase whether SBA and OBA effects would be impacted by a reward object. To test the SBA effect, the mean RTs were analyzed through a 2 (cue position: reward, no-reward) × 2 (cue validity: valid, invalid) repeated measures ANOVA (invalid condition was derived from the mean RTs of the ISO and IDO conditions.). While revealing a significant SBA effect with faster average RTs of 9.7 ms for valid targets compared to invalid targets (F (1, 35) = 30.97, *p* < 0.001, ηp^2^ = 0.47), a significant main effect of cue position was also observed (F (1, 35) = 6.81, *p* = 0.013, ηp^2^ = 0.16). More importantly, the two-way interaction between cue validity and cue position was significant (F (1, 35) = 5.11, *p* = 0.030, ηp^2^ = 0.13), with the SBA effect being more evident under the reward condition compared to the no-reward condition. To investigate whether the enhanced SBA effect in the reward compared to no reward condition was due to faster valid RTs or slower invalid RTs, we performed a detailed analysis of the effect of reward within each condition. And, we found that under the valid condition, the RTs in the reward condition (M = 320.35, SE = 4.98) were significantly faster than in the no-reward condition (M = 323.94, SE = 5.08, t (35) = −3.35, *p* = 0.003, Cohen‘s d = −0.12), but under the invalid condition, there was no significant difference (t (35) = 0.34, *p* = 0.733, Cohen’s d = 0.01), suggesting that the interaction between cue validity and cue position caused rewards to speed up RTs under valid conditions rather than slowing them down under invalid conditions.

Continually, we analyzed the influence of a reward object on the OBA effect by conducting a 2 (cue position: reward, no-reward) × 2 (cue validity: ISO, IDO) repeated measures ANOVA. The statistical analysis indicates that there was no significant main effect of cue position (F (1, 35) = 0.15, *p* = 0.70, ηp^2^ = 0.01). However, a significant main effect of cue validity was observed (F (1, 35) =89.91, *p* < 0.001, ηp^2^ = 0.72), which suggests a significant OBA effect. Similarly, the interaction between cue position and cue validity yields significant results (F (1, 35) = 4.13, *p* = 0.05, ηp^2^ = 0.11, see Figure 3b). The results suggested that a reward object also affects OBA. However, further post-hoc tests did not reveal significant differences between reward and no-reward conditions on either ISO or IDO. In Experiment 1, it became evident that the SBA effect was more pronounced in the reward condition as opposed to the no-reward condition. Moreover, when cues appeared on reward objects, a larger OBA effect was observed, implying that a reward object exerts an influence on SBA and OBA. However, in social interactions, we are confronted with more complex objects and stimuli, and whether this effect remains stable when stimulus complexity increases is yet to be examined. Also, a prior study aroused a question that simple geometric objects could not represent the patterns of selective attention in the real world [23,24]. To solve this problem, we conducted a replication study in Experiment 2 using real objects instead of colored rectangles. We anticipated obtaining similar results to ensure the stability and generalizability of the conclusions drawn in Experiment 1.

**Figure 3 behavsci-14-00505-f003:**
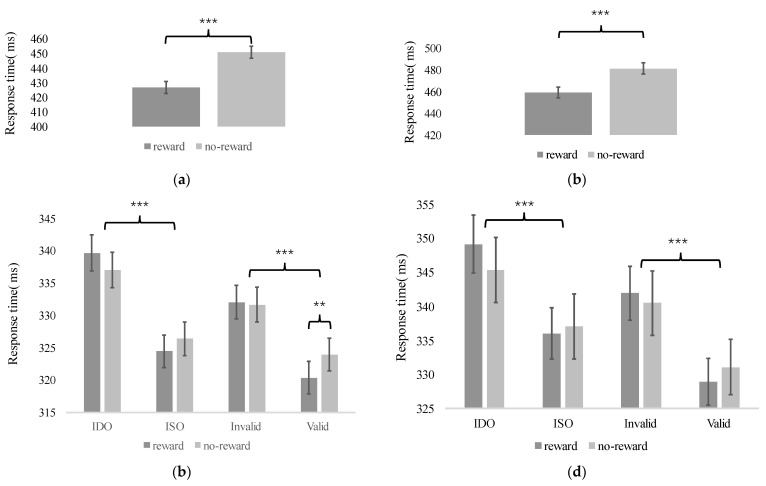
Results of Experiment 1 and 2. (**a**,**b**) were the training phase’s results of Experiment 1 and 2. (**c**,**d**) were the test phase’s results of Experiment 1 and 2. Asterisks indicate statistically significant differences in main and simple effects between conditions (** *p* < 0.01, *** *p* < 0.001). Non-significant comparison results are not marked in the figure (*p* < 0.05). Error bars represent standard errors of the mean.

#### 3.2.2. Experiment 2

RTs were analyzed in a 2 (cue position: reward, no-reward) × 2 (cue validity: valid, invalid) repeated measures ANOVA to examine the SBA effect. We observed a significant main effect of cue validity (F (1, 35) = 25.81, *p* < 0.001, ηp^2^ = 0.42), but there was no significant cue position effect (F (1, 35) = 0.05, *p* = 0.823, ηp^2^ < 0.01). Importantly, cue position effect significantly interacted with the cue validity effect (F (1, 29) = 4.96, *p* = 0.033, ηp^2^ = 0.124), which replicated the results of Experiment 1. Nevertheless, further tests did not reveal significant effects between the reward and no-reward conditions, neither in the valid condition (t (35) = −1.23, *p* = 0.446, Cohen’s d = −0.05), nor in the invalid condition (t (35) = 0.83, *p* = 0.446, Cohen’s d = 0.03), which differs from the results from Experiment 1. Also, we tested the effect of a reward object on OBA by using a 2 (cue position: reward, no-reward) × 2 (cue validity: ISO, IDO) repeated measures ANOVA. Confirming the presence of OBA, the results showed a significant main effect of cue validity (F (1, 35) = 28.92, *p* < 0.001, ηp^2^ = 0.45). However, the main effect of the cue position (F (1, 35) = 0.57, *p* = 0.456, ηp^2^ = 0.02) and the two-way interaction effect between cue position and cue validity were not significant (F (1, 35) = 3.50, *p* = 0.091, ηp^2^ = 0.09, see Figure 3d). For further details, please refer to Table 1.

Regarding the inconsistent results regarding the influence of a reward object on the OBA effect, we further integrated the two experiments and conducted a 2 (Experiment: Experiment 1, Experiment 2) × 2 (cue position: reward, no-reward) × 2 (cue validity: ISO, IDO) repeated measure ANOVA. The results revealed that the three-way interaction did not yield significant results (F (1, 70) < 0.001, *p* = 0.955, ηp^2^ < 0.001), while the two-way interaction between the cue validity and cue position was significant (F (1, 70) = 7.55, *p* = 0.008, ηp^2^ = 0.10). Additional replicated Bayesian repeated measures ANOVA results also provide positive evidence for the interaction between the OBA effect and reward object effect (BF_incl_ = 3.117) and strong evidence supporting the insignificance of the three-way interaction, indicating consistency in the patterns between the two experiments (BF_incl_ = 0.053) [25]. These results suggest that in Experiment 2, although the modulation of a reward on OBA was not significant, the pattern observed did not significantly differ from that of Experiment 1.

## 4. Discussion

Reward has been shown to influence selective attention, but it is unclear whether it is affected by a reward object. Thus, the present research was designed to address the impact of a reward object on SBA and OBA, utilizing a modified two-rectangle paradigm, in order to fill the current research gap and provide stronger empirical support for the VDAC theory. In the two experiments reported here, we found that the OBA effect and SBA effect could be obtained but modulated by a reward object.

As expected, we found that the SBA effect was influenced by a reward object. This finding, although different from the notion of SBA’s stability in previous research [3,26], demonstrates the influence of a reward object on SBA, still consistent with numerous studies on reward which have consistently shown that stimuli associated with rewards are accompanied by a stronger spatial bias [27,28]. However, what distinguishes this study from previous research is that, unlike the typical reward guiding spatial attention towards specific locations or features, in the two-rectangle paradigm used in this study, rewards were served as background information enhancing the cueing effect at valid locations. This demonstrates that besides directly guiding selective attention, a reward object can also indirectly participate in the process of guiding SBA through other factors, further supporting the relationship between a reward object and SBA. Additionally, it is worth noting that, given the paradigm used in this study where the target is more likely to appear under the cued condition, this probability might also induce stronger attention capture to the cued location as an endogenous cue. The significant SBA effect is, to some extent, related to this cue effect. Further studies could modify the paradigm to obtain a more accurate effect size.

Also, we demonstrated the modulating effect of OBA when using a reward object. This OBA effect was stronger when the cue appeared on a reward object compared to no reward object. Based on the VDAC theory, prior reward associations alter the saliency of objects [29]. Similarly, a substantial body of research has demonstrated convergence between reward-driven attention and the brain regions responsible for OBA [30,31]. Yamamoto et al. [32] demonstrated that rewards modulate object-selective responses in the caudate tail. Therefore, when a cue appears on a previously reward object, shifting attention to a no-reward object becomes more difficult; whereas, when the cue comes to a no reward object, the previously reward object, due to its heightened saliency, weakens the boundary constraints of the no-reward object, facilitating the transfer of attention to the other object, thus modulating the OBA effect. It is also important to recognize that, although in Experiment 2 the interactive effect of OBA and reward object yielded non-significant results, subsequent three-way ANOVA and Bayesian tests provided strong evidence supporting the lack of significant differences between Experiment 1 and Experiment 2, suggesting similar patterns between the two. Some may argue that this same pattern might also suggest the possibility of a false positive in the results of Experiment 1. However, our three-way ANOVA also found a significant interaction effect between the cue position and the cue validity, indicating that the results are more likely consistent with Experiment 1 rather than Experiment 2. Additionally, past research has found that the stability of object effects grows when real objects are used, making it more challenging for other factors to modulate OBA effects [22,33]. Therefore, the separation of experimental results should be attributed to an object’s complexity. In future studies, the effect of reward objects on real objects should be investigated by repeating the rewards multiple times or by increasing the reward amount.

The present finding contributes to an evolving literature of research examining the influence of a reward object on the deployment of selective attention. In many previous studies, reward for specific locations or features has been shown to distract attention or enhance it. However, this study found that this enhancing effect is also constrained by the boundaries of objects which suggests that the OBA effect does not simply occur as a default strategy [34], as previously thought. Instead, the reward attribute of an object can also participate in the process of object advantage, thus modulating selective attention. Furthermore, this study also provides theoretical underpinnings for the treatment of addictive behaviors. The incentive salience hypothesis proposed by Berridge and Robinson [35] suggests that behaviors and stimuli associated with rewards will prioritize cognitive executive control through dopamine secretion. However, the distinctiveness of a reward object from other types of rewards may lie in its ability to attract attention without requiring explicit conscious processing, even when the reward itself is unavailable [31,36]. This study first found that mere repeated exposure to objects previously associated with rewards would cause participants’ attention to shift, explaining the attentional performance of addicts when confronted with stimuli such as alcohol or drugs: once dopamine-associated stimuli appear, addicts will be uncontrollably guided by attention. Consequently, this discovery provides theoretical support for subsequent treatments of addiction and other mental illnesses.

Nevertheless, this study has some limitations. First, although this study only aimed to use reward and no-reward objects to explore the effects of reward objects rather than the differences between high and low reward attributes, previous research still suggests that comparing high and low rewards provides purer evidential results when investigating the effects of reward-induced attentional capture [18]. Future research can further explore the effects of reward objects by comparing high and low rewards. Additionally, while this study demonstrates the differences in attentional capture abilities between high-reward and low-reward objects, further research is needed to determine whether response times differ when two high-reward or two low-reward objects are presented simultaneously. This would help establish whether there is an overall activation effect of rewards in the current findings. Lastly, given our conclusions, the focus on the reward attributes of objects could provide theoretical support for the treatment of agnosia [37]. Further clinical research should explore the potential therapeutic effects of rewards on agnosia.

## 5. Conclusions

To conclude, this study examines the special response of selective attention to reward objects through two experiments based on a modified two-rectangle paradigm. One notable strength of our research is that it provides initial evidence that reward has a general modulating effect on selective attention, whether it is based on space or objects, and whether it involves simple or complex objects. These results strongly support the VDAC theory and provide further empirical evidence for it. When cues appear on reward objects, both spatial effects and object effects are enhanced, collectively forming priority maps that determine the final attentional selection.

## Figures and Tables

**Figure 1 behavsci-14-00505-f001:**
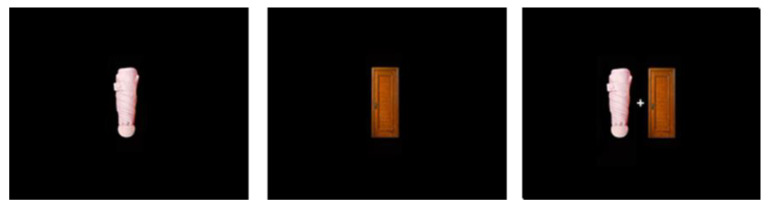
The stimulus used in Experiment 2.

**Figure 2 behavsci-14-00505-f002:**
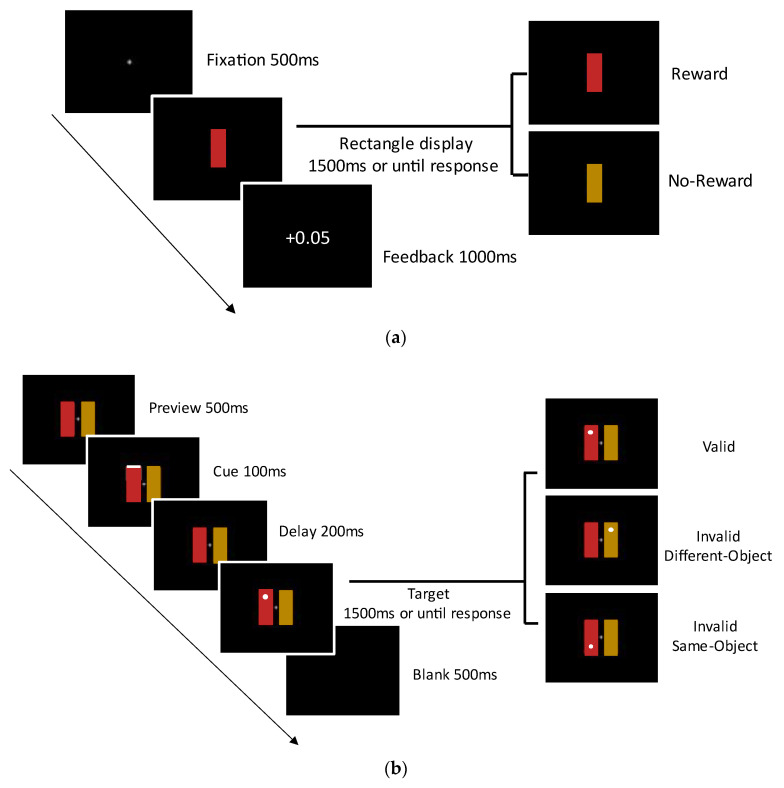
The stimuli used in Experiment 1. ‘Reward’ trial: the rectangle had 80% chance of being rewarded. ‘No-reward’ trial: the rectangle had no reward association. (**a**) Training Phase. (**b**) Test phase.

**Table 1 behavsci-14-00505-t001:** The mean RTs of test phase of Experiment 1 and Experiment 2. The error terms, in parentheses, reflect the within-subjects SEM.

		Valid	Invalid	IDO	ISO
Experiment 1	Reward	320.35 (4.97)	332.04 (5.21)	339.69 (5.51)	324.43 (5.11)
	No-reward	323.94 (5.08)	331.67 (5.32)	337.01 (5.54)	326.39 (5.19)
Experiment 2	Reward	328.96 (6.81)	341.98 (7.92)	349.19 (8.47)	336.02 (7.55)
	No-reward	331.11 (7.28)	340.52 (8.62)	345.40 (8.79)	337.06 (8.71)

## Data Availability

Data will be made available with reasonable request to the corresponding author.

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
