# Peer review of "The Impact of Reward Object on Object-Based Attention"

_behavsci, 2024, doi:10.3390/bs14060505_

Round 1

Reviewer 1 Report

Comments and Suggestions for Authors

The following paper describes a pair of experiments examining the effects of assigning reward value to previously neutral objects on space and object based attention. Results indicate that reward value effects both of these, at least for simple objects, but less clearly for real world objects. The design and description of the study are effective. I would note the following potential improvements for the next draft of the manuscript:

1. Pages 3-4, Lines 147-149:

It should be noted that by having the cue be predictive (50% accurate vs. 25% for 4 possible target locations) as well as location-based, means that both endogenous and exogenous attention may be at play here. This should be at least briefly noted in the discussion.

2. Figure 1:

Text should be larger for readability.

The “(a)” and “(b)” labels should be next to the section titles (i.e. “Training Session” and “Test Session”), or they should be omitted if not referenced anywhere in the figure caption or rest of the paper.

3. Page 7, Lines 249-263 (and in discussion):

“Marginally significant” should be rephrased. If the cutoff for significance is .05, then anything above that is simply not significant. It’s still appropriate to note the Bayesian comparison between the experiments with significant and non-significant results, but then it must be noted that this is just as likely to indicate a potential false positive in the results that achieved significance.

4. General Question/Possible discussion item:

Were there any trials where both rectangles are previously rewarded/unrewarded as a point of reference? Would this be considered in the future?

Comments on the Quality of English Language

There are English phrasing issues throughout, but they did not significantly impact understanding of the paper.

Reviewer 2 Report

Comments and Suggestions for Authors

It has been demonstrated that rewards affect selective attention; however, prior research has mostly examined rewards linked to particular places or characteristics, with little exploration of the effect of reward object on object-based attention (OBA). Consequently, it is still unknown whether items that were once linked to incentives have an impact on OBA. We used a paradigm that incorporated a modified two-rectangle paradigm with a reward training phase in two studies to address this problem. The findings show that reward objects influence both OBA and space-based attention (SBA). The SBA and OBA effects are stronger when cues are presented on reward objects as opposed to non-reward objects.

- It is very unusal practice that experiment 1 is written before the Methods section. Please see some published articles for the arrangment.

- In figure 2, the p-values are not clear to which bar they are saying something. Do point out them. 

- Which statistical tests were adopted by the authors, as the number of particpants seems low.

- Figure 1 is not clear. Use some fonts and figures that are understandble.

-Seperating both experiments and explaining them in seperate methods section seems no good, rather confuse the reader. 

- Whether there are some results of both experiments are correlated? to what extend. A seperate subsection is needed to elaborate this. 

- Conclusion section is weak, that needs improvment. 

- The authors must double check the consistency of references. 

- The authors would like to thoroughly read the article for correctness of sentences and typos. 

Round 2

Reviewer 2 Report

Comments and Suggestions for Authors

The authors have addressed my concerns